# Amaryllidaceae Alkaloids from *Clivia miniata* (Lindl.) Bosse (Amaryllidaceae): Isolation, Structural Elucidation, and Biological Activity

**DOI:** 10.3390/plants11223034

**Published:** 2022-11-10

**Authors:** Marcela Šafratová, Jana Křoustková, Negar Maafi, Daniela Suchánková, Rudolf Vrabec, Jakub Chlebek, Jiří Kuneš, Lubomír Opletal, Franz Bucar, Lucie Cahlíková

**Affiliations:** 1Department of Pharmacognosy and Pharmaceutical Botany, Faculty of Pharmacy, Charles University, Heyrovskeho 1203, 500 05 Hradec Kralove, Czech Republic; 2Department of Bioorganic and Organic Chemistry, Faculty of Pharmacy, Charles University, Heyrovskeho 1203, 500 05 Hradec Kralove, Czech Republic; 3Institute of Pharmaceutical Sciences, University of Graz, Beethovenstraße 8, 8010 Graz, Austria

**Keywords:** acetylcholinesterase, Amaryllidaceae, butyrylcholinesterase, *Clivia miniata*, clivimine B

## Abstract

*Clivia miniata* (Amaryllidaceae) is an herbaceous evergreen flowering plant that is endemic to South Africa and Swaziland and belongs to one of the top-10 traded medicinal plants in informal medicine markets in South Africa. The species has been reported as the most important component of a traditional healer’s pallet of healing plants. Eighteen known Amaryllidaceae alkaloids (AAs) of various structural types, and one undescribed alkaloid of homolycorine-type, named clivimine B (**3**), were isolated from *Clivia miniata*. The chemical structures of the isolated alkaloids were elucidated by a combination of MS, HRMS, 1D and 2D NMR techniques and by comparison with literature data. Compounds isolated in a sufficient quantity, and not tested previously, were evaluated for their in vitro acetylcholinesterase (AChE; E.C. 3.1.1.7) and butyrylcholinesterase (BuChE; E.C. 3.1.1.8) inhibition activities.

## 1. Introduction

Plants of the Amaryllidaceae family have a notable place in the history of traditional medicine and are known to produce structurally unique Amaryllidaceae alkaloids (AAs) with a wide range of biological activities including antiviral, antibacterial, antitumor, antimalarial, and cholinesterase inhibition properties [1,2,3,4]. The most-known compound within AAs is galanthamine, originally isolated from the bulbs of snowdrops; this compound is approved by the Food and Drug Administration (FDA) for the treatment of AD as a long-acting, selective, reversible, and competitive AChE inhibitor [5].

The genus *Clivia* Lindl. (Amaryllidaceae) is a small group of ornamental perennial plants native to southern Africa used by traditional healers in South Africa in the treatment of a number of ailments of a physiological or spiritual origin and has been implicated in human poisonings [6]. The most common species, *Clivia miniata*, commonly named as Natal lily, bush lily, or Kaffir lily, is a handsome, ornamental perennial herb usually cultivated for its beautiful flower [7]. It is traditionally used as an emetic remedy for fevers, as uterine tonics, and for the treatment of snakebites, barrenness, and urinary complaints [8]. Rasethe et al. reported in 2019 that the bulb of *C. miniata* has been used for the treatment of human immunodeficiency virus, arthritis, skin disorders, and tuberculosis by the people of Limpopo Province of South Africa [9]; Leven et al. stated its folkloric use due to its antiviral activity [10]. Recently, three new AAs containing 2,6-dimethylpyridine-, homolycorine- (cliniatine C), and galanthamine-parts (cliniatine A and cliniatine B) in their structure have been obtained from the whole plant [11]. Previously, similar alkaloids containing a 2,6-dimethylpyridine core, such as cliviamartine, miniatine, and clivimine, have been isolated only from *Clivia miniata* (Figure 1) [12,13,14].

Alzheimer’s disease (AD) is a multifactorial, neurodegenerative, progressive, and fatal disorder characterized by loss of cholinergic neurons in the hippocampus and cerebral cortex, mainly affecting cholinergic neurotransmission [15]. AD is the most common neurodegenerative disorder in the elderly, affecting around 11% of the population over the age of 65 and nearly half of people aged 85 years and older [16]. Based on the cholinergic hypothesis, a low production of acetylcholine (Ach) initiates AD. The first drugs approved for AD therapy were inhibitors of the enzyme acetylcholinesterase (AchE), which is responsible for the hydrolysis of Ach. Three AchE inhibitors, namely donepezil, galanthamine, and rivastigmine, are currently used as the main therapeutic option for AD treatment [17]. Since the Amaryllidaceae alkaloid galanthamine was introduced into clinical practice in 2001, other Amaryllidaceae alkaloids have received attention as potential cholinesterase inhibitors [18,19,20].

As a part of our ongoing research on Amaryllidaceae alkaloids with the potential to treat neurodegenerative diseases, together with the absence of a detailed complete phytochemical report on *C. miniata*, we were encouraged to examine this species. Isolated alkaloids that had not been previously studied for their activity against human AChE (*h*AChE) and human BuChE (*h*BuChE) were submitted for biological evaluation to reveal their inhibition potential.

## 2. Results and Discussion

### 2.1. Phytochemical Study of Clivia miniata

Extensive chromatographic purification led to the isolation of eighteen known and one previously undescribed AAs. The compounds were identified by a combination of MS, ESI-HRMS, 1D- and 2D-NMR experiments, and circular dichroism spectra (CD) and by comparison of the obtained data with the literature as: 1-O-acetylcaranine (1) [21], clivimine (2) [12], cliniatine C (4) [11], 4′-O-demethylbelladine (5) [20], caranine (6) [22], tazettine (7) [23], clivonine (8) [24], galanthamine (9) [25], haemanthamine (10) [24], vittatine (11) [26], haemanthidine (12) [27], hippeastrine (13) [28], nobilisitine B (14) [29], 3-O-acetyl-8-O-demethylmaritidine (15) [24], 11-hydroxyvittatine (16) [30], sternbergine (17) [31], 8-O-demethylmaritidine (18) [32], and lycorine (19) [27] (Figure 2). The isolated alkaloids belong to the lycorine (1, 6, 17, 19), homolycorine (2, 4, 8, 13, 14), belladine (5), tazettine (7), galanthamine (9), and haemanthamine (10*–*12, 15, 16, 18) structural types. The major alkaloids isolated were lycorine (19), clivimine (2), and haemanthamine (10); the remaining alkaloids were present in low or trace amounts. Alkaloids 1, 3, 5, 7, 9, 11, 12, and 14–17 were isolated for the first time from *Clivia miniata*.

The undescribed alkaloid **3**, named as clivimine B, was obtained as a white amorphous solid. A protonated molecule [M+H]^+^ at *m*/*z* 792.2761 was observed in the ESI-HRMS, corresponding to the molecular formula C_43_H_42_N_3_O_12_^+^ (calcd. 792.2763), which suggested an olefinic bond in a molecule similar to clivimine (possible fragmentation of compound **3** is indicated in the mass spectral data in Appendix A). This assumption was later confirmed by a thorough NMR analysis, starting with 1D NMR experiments. A characteristic core of 2,6-dimethylpyridine-3,5-dicarboxylate was identified in the ^1^H NMR spectrum, represented by a deshielded singlet of H-4″ resonating at 8.84 ppm and two singlets of methyl groups H-7″ and H-10″ at 3.01 and 3.00 ppm. As for the remaining units in this molecule, the aromatic region of the ^1^H NMR spectrum revealed a further four singlets belonging to two 1,2,4,5-tetrasubstituted benzene rings (δ_H_ 7.84, s, H-8′; 7.78, s, H-11; 7.66, s, H-8; 7.45, s, H-11′) substituted with dioxomethylene groups, which were recognized from the characteristic overlapping doublets of such groups (δ_H_ 6.23, d, overlap, *J* = 6.5 Hz and 6.23, d, overlap, *J* = 6.5 Hz for H-13′; 6.14, d, overlap, *J* = 3.9 Hz and 6.14, d, overlap, *J* = 3.9 Hz for H-13). Furthermore, three deshielded signals were considered to belong to OCH groups (δ_H_ 6.20, dd, *J* = 8.4 Hz, *J* = 6.2 Hz, H-5′; 5.83, ddd, *J* = 3.1 Hz, *J* = 3.1 Hz, *J* = 2.9 Hz, H-5; 4.38, dd, *J* = 12.7 Hz, *J* = 2.9 Hz, H-5a) and two singlets of two NCH_3_ groups were recognized at 2.33 and 2.30 ppm. The rest of the signals in the aliphatic region were not that specific, with another five sp^3^-methines and six sp^3^-methylenes. The ^13^C NMR spectrum revealed 43 carbons. All protons were unambiguously assigned to corresponding carbons in the HSQC experiment. The structure of **3** was divided into three parts—units A, B, and C—for its structure elucidation employing 2D NMR data (see Figure 3).

Starting with unit A, COSY data revealed spin systems of H-2/H-3 and H-4/H-5/H-5a/H-11b/H-11c, depicted in Figure 3 by blue-colored bonds. Because of overlapping proton resonances at 2.44–2.31 ppm, the H2BC experiment was the key to the determination of the CH group at position 3a. Interactions of H-3 (δ_H_ 2.06–1.97, m), H-4 (δ_H_ 1.93–1.89, m), and H-11c (δ_H_ 2.83, dd, *J* = 10.0 Hz, *J* = 6.5 Hz) with C-3a (δ_C_ 32.8) were found in the H2BC spectrum. Thus, an octahydroindole moiety was identified, and then an attachment to one of the 1,3-benzodioxole rings was recognized by HMBC correlations of H-11/C-11b and H-11b/C-11. Long-range correlations of H-8 to C-7 and H-5a to C-11a revealed a lactone in unit A, leading to closure of a homolycorine type of scaffold.

Only two small spin systems, H-2′/H-3′ and H-4′/H-5′, were identified in unit B (see Figure 3). The conjunction of H-3′/H-3′a/H-4′ was covered by an overlap at 2.44–2.31 ppm, as previously mentioned in unit A. Linkage of C-3′/C-3′a/C-4′ and C-3′a/C-11′c was identified by H2BC correlations. The double bond (δ_C_ 149.7, C-5′a; 112.6, C-11′b), differentiating this alkaloid from clivimine, was identified by the HMBC correlations from H-4′ and H-11′c to C-5′a and then H-5′ and H-11′c to C-11′b. Thus, an HMBC correlation from H-11′ to C-11′b showed a connection of this described hexahydroindole with the remaining 1,3-benzodioxole moiety. The last substituent of the benzene ring, a carbonyl with δ_C_ 160.9, was determined by HMBC correlation H-8′/C-7′ and by the chemical shift of C-5′a, suggesting an attachment to the electronegative oxygen belonging to the respective lactone.

Due to very similar ^1^H and ^13^C NMR resonances at positions 7″/10″, 2″/6″, and 3″/5″, methyl groups and respective quaternary sp^2^-carbons of the pyridine ring were not distinguished to one particular position (see Table 1). However, carboxylic carbons were assigned by HMBC interaction from unit B (H-5′/C-9″); interestingly, no correlation was observed for H-5/C-8″.

Since compound **3** has eight chiral carbons, determining the stereochemistry was the next step in elucidating the chemical structure. Unfortunately, attempts to prepare a sample for single-crystal X-ray diffraction were in vain. Even so, the relative configuration for unit A and separately for unit B was successfully determined, based on NOESY interactions and spin–spin couplings, and, therefore, equally important was the comparison of NMR data with structurally related compounds for which the stereochemistry was thoroughly established. Thoroughly is an important adverb at this point because many errors have been made in the stereochemistry of alkaloids with a homolycorine skeleton, as summarized by, e.g., Wang and co-workers [33] in their work focused on the total synthesis of (±)-clivonine.

Concerning the stereochemical evidence in unit A (see Figure 4), a doublet of doublets (*J* = 12.7 Hz and 2.9 Hz, H-5a) showed a through-space NOESY interaction with H-5, suggesting an axial–equatorial or diequatorial relationship together with their *J*-coupling of 2.9 Hz. The axial orientation of H-5a was subsequently confirmed by the large *J*-coupling of 12.7 Hz shared with H-11b, which defined the *trans*-orientation of ring-junction hydrogens H-5a/H-11b and thus defined the axial–equatorial orientation for H-5/H-5a. In addition, the other vicinal 10.0 Hz spin–spin interaction of H-11b arose from the adjacent H-11c, which determined their *trans*-orientation. The position of H-11c, supported by NOESY correlation with H-5a, suggested that it faces the same side of the molecule. The remaining stereocenter H-3a was identified by NOESY interaction with H-11c, indicating a *cis*-orientation of these hydrogens. Afterwards, cliniatine C [11], with the absolute configuration determined, served as a comparison of NMR data (see Appendix A). No significant difference in chemical shifts or *J*-couplings was found, which supported the established (3a*R*^*^,5*S*^*^,5a*R*^*^,11b*S*^*^,11c*R*^*^)-configuration of unit A.

Regarding the three remaining chiral centers in unit B of **3**, a doublet with *J*-coupling of 5.3 Hz representing H-11′c showed a through-space correlation with H-3′a, indicating their location on the same side of the molecule, as in the corresponding *cis-*oriented CH groups of unit A (see Figure 5). NOESY correlation was observed for H-3′a and H-5′, suggesting a pseudoaxial orientation of allylic H-5′. Determination of H-5′ orientation was supported by comparison with related structures described in the total synthesis of narseronine [34] that were analyzed, among other techniques, by X-ray crystallography. C5-epimers were perfect comparison targets (see Appendix A). Thus, the relative configuration of unit B was elucidated as (3ʹa*R*^*^,5ʹ*S*^*^,11ʹc*R*^*^).

Although nobilisitine B (**14**), another alkaloid isolated in our phytochemical work, had already been described by Evidente et al. [29], we would like to emphasize that our NMR data were consistent with those reported, except for the assignments of C-7a, C-9, C-10, and C-11a where the resonances were apparently swapped. The corrections are the following: δ_C_ 118.7, C-7a; 146.7, C-9; 155.6, C-10; and 139.9, C-11a, which correlate with the electron distribution on the benzene ring with such substitution. Note that this alkaloid is not a derivative of nobilisitine A. It is tempting to assume this from the name, but it is a 5-*O*-(3-hydroxybutanoyl) derivative of clivonine. Notably, the path to elucidating the true absolute configuration of nobilisitine A was quite torturous. First, Schwartz and co-workers [35] published its synthesis with the originally proposed stereochemistry. As it turned out, their NMR data differed from those of the isolated alkaloid from *Clivia nobilis*. Lodewyk and Tantillo [36], therefore, followed up with the NMR data calculations for 8 out of 16 possible diastereomers of nobilistine A and predicted the thermodynamically most favorable configuration, which was then confirmed by Schwartz et al. [37] in the final revisions of nobilisitine A stereochemistry (see Figure 6). This alkaloid was found only in traces in the studied extract.

### 2.2. Biological Activity of Isolated Amaryllidaceae Alkaloids

All the isolated alkaloids that had not been tested previously for their *h*AChE/*h*BuChE inhibition potencies, and which were obtained in sufficient amounts, were characterized for their activities. Galanthamine and eserine were used as reference compounds. Compounds with IC_50_ values higher than 100 µM in the *h*AChE/*h*BuChE inhibition tests were considered as inactive. The results, expressed as IC_50_ values, are summarized in Table 2. All alkaloids studied within the current study showed only weak *h*AChE/*h*BUChE inhibition potency (IC_50_ > 100 µM). Low *h*AChE/*h*BUChE inhibitory activities are not surprising since most of the newly tested alkaloids belong to the homolycorine structure type and these substances generally demonstrate only low cholinesterase inhibition activities. When we compare structurally close lycorine-type alkaloids, the reason behind the activity of some lycorine-type alkaloids could be the presence of free hydroxyl groups in positions C1 and C2, which are not present in those of the homolycorine type. These hydroxyls allow the presence of additional functional groups that improve binding in the active site of *h*AChE/*h*BuChE (e.g., 1-*O*-acetyllycorine). Nevertheless, the vast majority of the lycorine alkaloids, including lycorine itself, are not significantly active against AChE and BuChE as well. The remaining isolated alkaloids have been screened already for their biological profile within our previous studies on Amaryllidaceae plants and are reported in the following literature [4,19,38,39,40].

## 3. Conclusions

In conclusion, the phytochemical investigation of the alkaloidal extract of *Clivia miniata* allowed the isolation of eighteen known AAs and one previously undescribed AA of different structural types. Compounds isolated in sufficient amounts and not studied before were assayed for their *h*AChE/*h*BuChE inhibition activity. Unfortunately, none of the alkaloids tested showed significant potential for inhibiting these enzymes. However, on the other hand, since lycorine is an attractive natural compound active in a very low concentration and with high specificity against a number of cancer cell lines, both in vivo and in vitro, and against various drug-resistant cancer cells, *Clivia miniata* can be recognized as a rich source of this attractive alkaloid.

## 4. Materials and Methods

### 4.1. General Experimental Procedures

All solvents were treated by using standard techniques before use. All reagents and catalysts were purchased from a commercial source (Sigma Aldrich, Czech Republic) and used without further purification. The NMR spectra were obtained in C_5_D_5_N at ambient temperature on a Varian VNMR S500 spectrometer. The chemical shifts were recorded as *δ* values in parts per million (ppm) and were indirectly referenced to tetramethylsilane (TMS) via the residual solvent signal (C_5_D_5_N—7.19, 7.55, and 8.71 ppm for ^1^H and 123.5, 135.5, and 149.9 ppm for ^13^C). Coupling constants (*J*) are given in Hz. The EI-MS were obtained on an Agilent 7890A GC 5975 inert MSD operating in EI mode at 70 eV (Agilent Technologies, Santa Clara, CA, USA). A DB-5 column (30 m × 0.25 mm × 0.25 μm, Agilent Technologies, USA) was used with a temperature program: 100–180 °C at 15 °C/min, 1 min hold at 180 °C, 180–300 °C at 5 °C/min, and 5 min hold at 300 °C; the detection range was *m*/*z* 40–600. The injector temperature was 280 °C. The flow-rate of carrier gas (helium) was 0.8 mL/min. A split ratio of 1:15 was used. ESI-HRMS were obtained with a Waters Synapt G2-Si hybrid mass analyzer of a quadrupole-time-of-flight (Q-TOF) type, coupled to a Waters Acquity I-Class UHPLC system. TLC was carried out on Merck precoated silica gel 60 F254 plates. Compounds on the plate were observed under UV light (254 and 366 nm) and visualized by spraying with Dragendorff’s reagent.

### 4.2. Plant Material

*Clivia miniata* was planted in the greenhouse of the Faculty of Pharmacy in Hradec Králové in 2018 and collected in May 2021. Botanical identification was performed by Prof. L. Opletal. A voucher specimen is deposited in the Herbarium of the Faculty of Pharmacy in Hradec Králové under number CUFPH-16130/AL-704.

### 4.3. Extraction and Isolation of Alkaloids

Freshly chopped whole plant (7 kg) was exhaustively extracted with ethanol (EtOH) (96%, v/v, 3×) by boiling for 30 min under a reflux condenser; the combined extract was filtered and evaporated to dryness under reduced pressure. The crude extract (45 g) was acidified to pH 1–2 with 5% hydrochloric acid (HCl; 1 L) and the volume of the suspension was made up to 4 L by water. The suspension was filtered; the filtrate was alkalized to pH 9–10 with a 10% solution of Na_2_CO_3_ and extracted with CHCl_3_ (3 × 5 L). The organic layer was evaporated to give 22 g of dark brown fluid residue. This alkaloid summary extract was again dissolved in 2% HCl (5 L), defatted with Et_2_O (3 × 2 L), and alkalized to pH 9–10 with 10% Na_2_CO_3_. The water layer was extracted with EtOAc (4 × 1 L), and CHCl_3_ (2 × 1 L). Both Dragendorff positive parts were evaporated and pooled together. A concentrated alkaloid extract (9.5 g) in the form of a brown syrup was obtained.

The alkaloid extract was further fractionated by Flash chromatography on the pre-packed silica gel cartridge (120 g). Light petrol (solvent A), EtOAc (solvent B), and MeOH (solvent C) were used for separation. First, the column was equilibrated with solvent A and solvent B in the ratio 3:2. The extract was separated using a 110 min program: solvent A and solvent B (3:2) were held for 10 min (0–10 min), then linearly increased to 100% B over 40 min (10–50 min), 100% B was held for 10 min (50–60 min), it was linearly increased over 40 min to 100% C (60–100 min), and 100% C was held for 10 min (100–110 min). Finally, the column was washed with 1% AcOH in MeOH (15 min). The flow rate was 60 mL/min and individual fractions were collected by UV monitoring the eluting analytes at 254 nm, 280 nm, 290 nm, and 308 nm. Fractions were collected by 25 mL and 50 mL, respectively, and monitored by TLC. Finally, 214 fractions were collected, combined into 18 fractions, and analyzed by GC-MS. Fractions with similar profiles were pooled together to give eight final fractions (**I**–**VII**).

Fraction **I** (105 mg) was treated by preparative TLC (cHx:Et_2_NH, 95:5, 2×) to give 1-*O*-acetylcaranine (**1**, 8 mg).

Fraction **II** (958 mg) was recrystallized from a mixture of EtOH/CHCl_3_ (1:1) to give clivimine (**2**, 348 mg). The mother liquor of fraction **II** was evaporated to obtain 562 mg residue, which was separated by preparative TLC (cHx:To:Et_2_NH, 60:35:5) to give clivimine B (**3**, 25 mg), cliniatine C (**4**, 2 mg), and 4′-*O*-demethylbelladine (**5**, 1.5 mg).

Fraction **III** (243 mg) was separated by preparative TLC (cHx:To:Et_2_NH, 50:45:5) to give two subfractions **IIIa**-**b**. Subfraction **IIIa** (156 mg) was separated by preparative TLC (cHx:To:Et_2_NH, 50:45:5) to give caranine (**6**, 12 mg) and subfraction **IIIa-1** (80 mg), which gave crystals of tazettine (**7**, 25 mg) from a mixture of EtOH/CHCl_3_ (1:1). The mother liquor of subfraction **IIIa-1** was evaporated and further chromatographed by preparative TLC (cHx:To:Et_2_NH, 65:30:5, 2×) to yield additional tazettine (8 mg) and clivonine (**8**, 4.5 mg). An acetone solution of subfraction **IIIb** yielded crystals of galanthamine (**9**, 15 mg).

Fraction **IV** (567 mg) yielded haemanthamine (**10**, 253 mg) from a mixture of EtOH/CHCl_3_ (1:1) after being twice recrystallized.

Fraction **V** (631 mg) was treated by preparative TLC (To:Et_2_NH, 95:5) to give further haemanthamine (86 mg) and subfraction **Va** (426 mg). Subfraction **Va** was repetitively separated by preparative TLC (To:cHx:Et_2_NH, 50:45:5, 2×) to give vittatine (**11**, 36 mg) and haemanthidine (**12**, 65 mg).

Fraction **VI** (198 mg) was subjected to preparative TLC (To:Et_2_NH, 9:1, 2×) to give hippeastrine (**13**, 15 mg) and subfraction **VIa** (28 mg). Subfraction **VIa** was repetitively separated by preparative TLC (cHx:To:Et_2_NH, 55:40:5, 2×) to give nobilisitine B (**14**, 1.5 mg) and 3-*O*-acetyl-8-*O*-demethylmaritidine (**15**, 4 mg).

Fraction **VII** (268 mg) was subjected to preparative TLC (To:Et_2_NH, 9:1, 2×) to give 11-hydroxyvittatine (**16**, 46 mg) and subfraction **VIIa** (137 mg). Subfraction **VIIa** was repetitively separated by preparative TLC (cHx:To:Et_2_NH, 10:85:5, 2×) to give sternbergine (**17**, 57 mg) and 8-*O*-demethylmaritidine (**18**, 6 mg).

Fraction **VIII** (4.5 g) gave lycorine (3.56 g), which was twice crystallized from a mixture of EtOH/CHCl_3_ (9:1).

### 4.4. Inhibition of hAChE and hBuChE

The activity of isolated alkaloids for the inhibition of human cholinesterases was assessed using the modified version of Ellman’s method [41], described recently by our group [18]. The detailed description of the assay can be found in the Appendix A.

## Figures and Tables

**Figure 1 plants-11-03034-f001:**
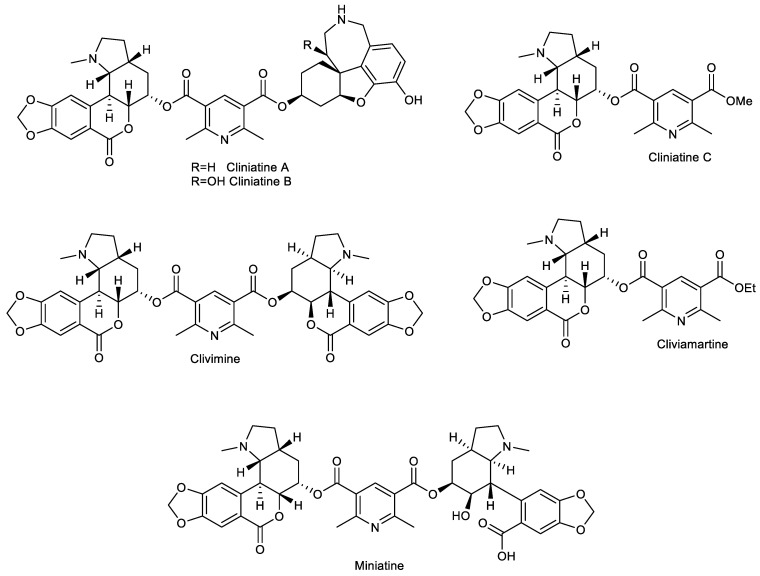
Amaryllidaceae alkaloids containing a 2,6-dimethylpyridine-scaffold isolated from *Clivia miniata*.

**Figure 2 plants-11-03034-f002:**
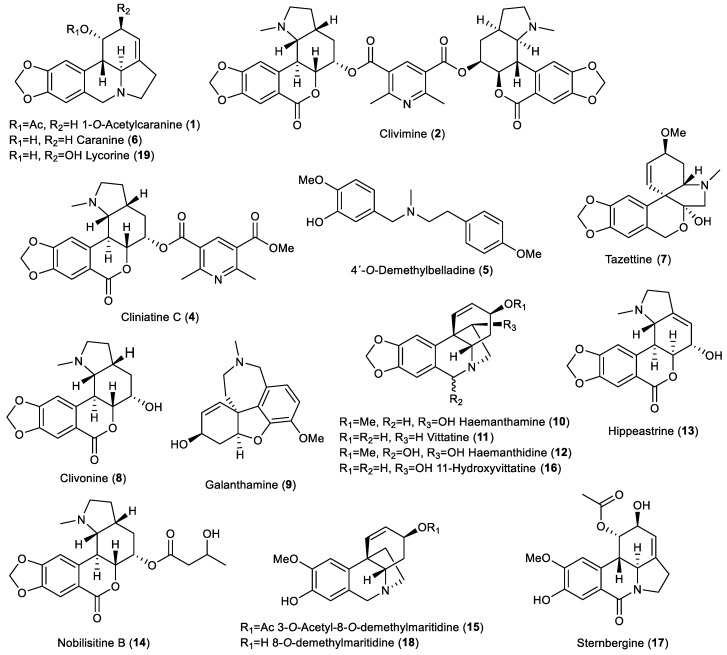
Isolated Amaryllidaceae alkaloids from *Clivia miniate* within current study.

**Figure 3 plants-11-03034-f003:**
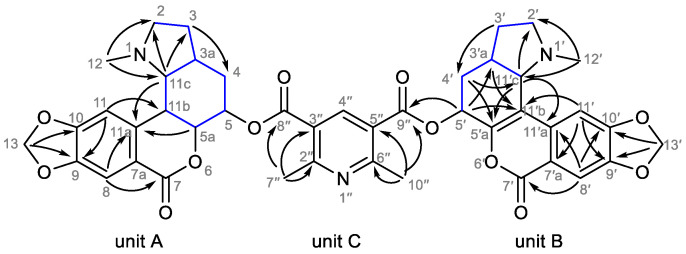
Key 2D NMR correlations for **3**—HMBC (arrows), COSY, and H2BC (blue bonds).

**Figure 4 plants-11-03034-f004:**
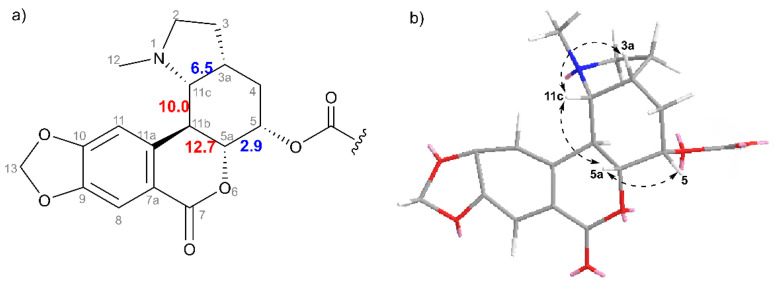
Determined relative configuration of unit A (**3**). (**a**) Values of *J*-coupling (given in Hz): highlighted in red represent the *trans*-orientation of coupling partners, while the blue color symbolizes the *cis*-orientation. (**b**) Stick presentation of unit A with NOESY correlations shown by dashed arrows.

**Figure 5 plants-11-03034-f005:**
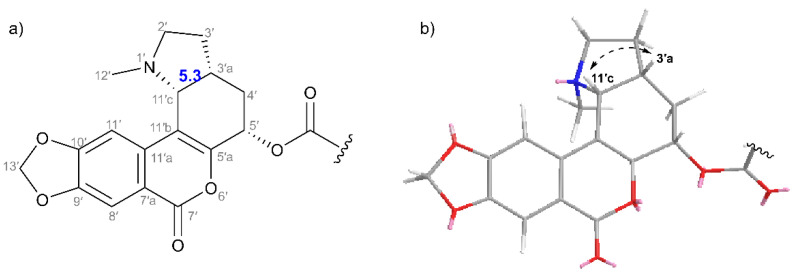
Determined relative configuration of unit B (**3**). (**a**) H-3′a/H-11′c value of *J*-coupling (given in Hz) determined the *cis*-orientation. (**b**) Stick presentation of unit B with NOESY correlation shown by dashed arrow.

**Figure 6 plants-11-03034-f006:**
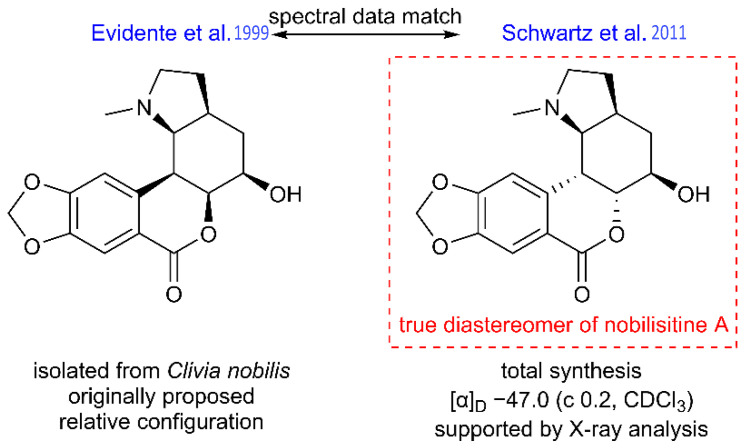
Nobilisitine A and its revised structure [29,37].

**Table 1 plants-11-03034-t001:** ^1^H NMR (499.9 MHz) and ^13^C NMR (125.7 MHz) data for **3** analyzed in C_5_D_5_N (*δ* in ppm and *J* in Hz).

Position	3		
	*δ* _H_	*δ* _C_	Type
2	3.46–3.38, m; 2.57–2.46, m	52.5	CH_2_
3	2.08–1.97, m	30.8	CH_2_
3a	2.44–2.35, m	32.8	CH
4	2.44–2.35, m; 1.93–1.89, m	27.8	CH_2_
5	5.83, ddd (3.1, 3.1, 2.9)	70.5	CH
5a	4.38, dd (12.7, 2.9)	79.0	CH
7		164.2	C=O
7a		119.4	C
8	7.66, s	109.1	CH
9		147.2	C
10		152.8	C
11	7.78, s	107.6	CH
11a		141.0	C
11b	3.39, dd (12.7, 10.0)	34.9	CH
11c	2.83, dd (10.0, 6.5)	69.6	CH
12	2.33, s	44.5	CH_3_
13	6.14, d, overlap (3.9); 6.14, d, overlap (3.9)	102.6	CH_2_
2′	3.07–2.98, m; 2.57–2.46, m	54.4	CH_2_
3′	1.97–1.93, m; 1.53–1.44, m	29.0	CH_2_
3′a	2.44–2.35, m	36.1	CH
4′	2.35–2.28, m; 2.14–2.08, m	32.2	CH_2_
5′	6.20, dd (8.4, 6.2)	69.3	CH
5′a		149.7	C
7′		160.9	C=O
7′a		116.5	C
8′	7.84, s	107.4	CH
9′		148.9	C
10′		154.1	C
11′	7.45, s	103.7	CH
11′a		135.2	C
11′b		112.6	C
11′c	3.57, d (5.3)	60.8	CH
12′	2.30, s	41.1	CH_3_
13′	6.23, d, overlap (6.5); 6.23, d, overlap (6.5)	103.3	CH_2_
2″		163.41 or 163.38	C
3″		122.78 or 122.53	C
4″	8.84, s	140.5	CH
5″		122.78 or 122.53	C
6″		163.41 or 163.38	C
7″	3.01, s or 3.00, s	25.33 or 25.29	CH_3_
8″		164.7	C
9″		165.1	C
10″	3.01, s or 3.00, s	25.33 or 25.29	CH_3_

**Table 2 plants-11-03034-t002:** In vitro results of *h*AChE and *h*BuChE inhibition assays of previously untested AAs isolated from *Clivia miniata*.

Compound	% Inhibition *h*AChE ± SEM^*a*^	IC_50_ *h*AChE ± SEM (µM) *^b^*	% Inhibition *h*BuChE ± SEM *^a^*	IC_50_ *h*BuChE ± SEM (µM) *^b^*
clivimine (**2**)	9.62 ± 1.75	>100	15.22 ± 8.23	>100
clivimine B (**3**)	10.09 ± 5.37	>100	7.69 ± 2.56	>100
cliniatine C (**4**)	5.72 ± 0.53	>100	9.43 ± 1.23	>100
clivonine (**6**)	34.43 ± 1.68	>100	28.22 ± 6.63	>100
3-*O*-acetyl-8-*O*-demethylmaritidine (**12**)	12.35 ± 1.29	>100	10.31 ± 2.06	>100
nobilisitine B (**14**)	40.03 ± 1.13	>100	12.57 ± 0.53	>100
sternbergine (**17**)	1.25 ± 0.23	>100	23.50 ± 2.28	>100
galanthamine *^c^*	94.88 ± 0.43	2.01 ± 0.14	68.23 ± 1.24	29.31 ± 3.49
eserine *^c^*	99.98 ± 0.02	0.20 ± 0.0.01	99.78 ± 0.04	0.30 ± 0.01

*^a^* Tested at 100 µM compound concentration; *^b^* Compound concentration required to decrease enzyme activity by 50%; the values are the mean ± SEM of three independent measurements, each performed in triplicate; *^c^* Reference compound.

## Data Availability

All data in this study can be found in the manuscript or in the Appendix A.

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
