# Peer review of "Amaryllidaceae Alkaloids from Clivia miniata (Lindl.) Bosse (Amaryllidaceae): Isolation, Structural Elucidation, and Biological Activity"

_plants, 2022, doi:10.3390/plants11223034_

Round 1

Reviewer 1 Report

The manuscript describes novel and interesting work on the isolation, structural elucidation and biological activity of alkaloids from Clivia miniata. The topic is of high interest for the readers of Plants. The phytochemical investigations are robust and thorough, being accomplished by the means of high performance techniques (ESI-HRMS, 1D- and 2D-NMR, CD). I would like to emphasize the rigour in elucidating the true absolute configuration of isolated alkaloids and also the novel structural information regarding nobilisitine B. The manuscript has a high scientific level and, at the same time, it is clearly written and comprehensible.

I recommend acceptance in the present form.

Author Response

Plants

Dear Editor,

We have carefully revised manuscript titled " Amaryllidaceae alkaloids from Clivia miniata (Lindl.) Bosse (Amaryllidaceae): isolation, structural elucidation, biological activity" Manuscript ID: plants- 2021762, according to the reviewers' suggestions. All changes are highlighted in red.

Comments from the reviewers:

Reviewer #1:

The manuscript describes novel and interesting work on the isolation, structural elucidation and biological activity of alkaloids from Clivia miniata. The topic is of high interest for the readers of Plants. The phytochemical investigations are robust and thorough, being accomplished by the means of high performance techniques (ESI-HRMS, 1D- and 2D-NMR, CD). I would like to emphasize the rigour in elucidating the true absolute configuration of isolated alkaloids and also the novel structural information regarding nobilisitine B. The manuscript has a high scientific level and, at the same time, it is clearly written and comprehensible.

I recommend acceptance in the present form.

Thank you very much for this evaluation of our work.

Reviewer #2:

Amaryllidaceae alkaloids from Clivia miniata (Lindl.) Bosse (Amaryllidaceae): isolation, structural elucidation, biological activity (by Šafratová et al.)

Nineteen alkaloids, including the novel compound 3, were described from a whole-plant EtOH extract of Clivia miniata. Structure elucidations were made via comparisons to the literature as well as by 1 and 2D NMR spectroscopic measurements. Biological evaluations were carried out against the AChE and BuChE enzymes. This is an interesting, well-researched and nicely written effort. Minor suggestions are made below which the authors could consider as a revision to the text.

Keywords: Arrange in alphabetic fashion

Revised.

P1L17. No need to use systematic names here as it has been used in the title.

Systematic name removed.

P3L96. Were there any other diagnostic ion fragments detected in the MS of compound 3? Were any IR spectra done? Did the authors run any UV spectra?

Possible fragmentation of compound 3 is indicated in the mass spectrum in the Supplementary material. But because our MS experiment gave only the HRMS spectrum, we are not able to study fragmentation mechanisms in more detail using MS/MS or MSn experiments.We did´nt measured IR and UV spectrum, whereas we have detailed 1D-, 2D-NMR experiments that give much more detailed informations about the structure of the new alkaloid.

P8. Low AChE and BuChE inhibitory activities were observed for all of the homolycorine compounds screened. Could any comparisons be made to the lycorine alkaloids, some of which are known for their potent activities against AChE?

When we compare structurally close lycorine-type alkaloids, the reason behind the activity of some lycorine alkaloids could be the presence of free hydroxyl groups of lycorine, which are not present in homolycorine. These hydroxyls allow the presence of additional functional groups that improve binding in the active site of AChE/BuChE (e.g., 1-O-acetyllycorine). Nevertheless, the vast majority of the lycorine alkaloids, including lycorine itself, are not significantly active against AChE/BuChE as well.

This explanation has been incorporated into the manuscript.

P9L265. Is boiling of the extract necessary?

Yes, it increases the yield of secondary metabolites from the plant material. At the same time, this method reduces the time-consuming extraction compared to classic maceration at room temperature.

Thank you very much for your valuable feedback and time spent reviewing the manuscript.

In light of these changes, we are positive that our revised manuscript meets the criteria to be published in Plants and would be of interest for all readers from the scientific community.

Lucie Cahlikova

Lucie Cahlíková, Prof.

Charles University, Prague

Faculty of Pharmacy, Hradec Králové

Department of Pharmacognosy and Pharmaceutical Botany

Hradec Králové, 4th, November, 2022

Reviewer 2 Report

Amaryllidaceae alkaloids from Clivia miniata (Lindl.) Bosse (Amaryllidaceae): isolation, structural elucidation, biological activity (by Šafratová et al.)

Nineteen alkaloids, including the novel compound 3, were described from a whole-plant EtOH extract of Clivia miniata. Structure elucidations were made via comparisons to the literature as well as by 1 and 2D NMR spectroscopic measurements. Biological evaluations were carried out against the AChE and BuChE enzymes. This is an interesting, well-researched and nicely written effort. Minor suggestions are made below which the authors could consider as a revision to the text.

Keywords: Arrange in alphabetic fashion

P1L17. No need to use systematic names here as it has been used in the title.

P3L96. Were there any other diagnostic ion fragments detected in the MS of compound 3? Were any IR spectra done? Did the authors run any UV spectra?

P8. Low AChE and BuChE inhibitory activities were observed for all of the homolycorine compounds screened. Could any comparisons be made to the lycorine alkaloids, some of which are known for their potent activities against AChE?

P9L265. Is boiling of the extract necessary?

Author Response

(The authors gave the same response as above.)
